# Synthesis and Characterization of Polymeric Blends Containing Polysulfone Based on Cyclic Bisphenol

**DOI:** 10.3390/polym14153148

**Published:** 2022-08-02

**Authors:** Mateusz Gargol, Beata Podkościelna

**Affiliations:** Department of Polymer Chemistry, Institute of Chemical Science, Faculty of Chemistry, Maria Curie-Skłodowska University, M. Curie-Skłodowska Sq. 3, PL-20031 Lublin, Poland; beata.podkoscielna@mail.umcs.pl

**Keywords:** polysulfone, acrylic resins, polymeric blends, photocuring, thermogravimetry, DMA analysis

## Abstract

The elaboration of the composition and methods of preparation of new types of materials is an important issue from the plastics industry’s point of view. The paper presents the polysulfone synthesis based on 4,4′-cyclohexylidenebisphenol (bisphenol Z). This compound was used (in an amount of 5 or 10 wt.% sample) for the synthesis and characterization of new polymeric blends based on the two different acrylic resins (EB-150 and EB-600) and the active solvent N-vinyl-2-pyrrolidone (NVP). The weight ratio of the used resin to solvent was 1:2; 1:1 or 2:1. These new materials were obtained applying the photoinitiated free radical polymerization with 2,2-dimethoxy-2-phenyloacetophenone as a photoinitiator used in an amount of 1 wt.%. Six polymeric blends and six copolymers without polysulfone were cured by this method. By means of ATR/FT-IR (Attenuated Total Reflection–Fourier Transform Infrared) spectroscopy the chemical structure of the synthesized polysulfone was proved. The effect of the presence of the polysulfone presence on the thermal properties of the obtained blends was analyzed by means of thermogravimetry and differential thermogravimetry (TG/DTG), as well as differential scanning calorimetry (DSC). Moreover, the dynamic mechanical studies (DMA) of these materials were also carried out, demonstrating which of the materials showed the influence of the percentage of polysulfone on the selected properties in the blended- and parent-copolymers samples.

## 1. Introduction

Polysulfones are amorphous and thermoplastic polymers. They are characterized by excellent thermal properties, even at high temperatures. Their glass transition temperature is in the range: 180–250 °C and distribution of them is between 400 and 550 °C, which is related to the presence of diphenyl ether and diphenyl sulfone units in their structures [1,2]. The most commonly used manner of synthesis of these compounds is condensing the dipotassium salts (e.g., 2,2-bis (p-hydroxyphenyl) propane bisphenol A) with 4,4’-bis (chlorophenyl) sulfone (DCDPS) [3]. It is also possible to use other compounds (different bisphenols) for the synthesis of different variants of polysulfones. By the above mentioned method, Butuc et al. prepared copolyether sulfones by a condensation reaction of DCDPS with bisphenol A and dipotassium salts of biphenols containing cycloaliphatic groups (cyclopentyl, cyclohexyl and cycloheptyl) in the presence of 4,4′-diphenyl disulfonyl chloride (DMSO) and anhydrous potassium carbonate [4]. The scientists compared the influence of the presence and size of the cycloalkanone ring on the solubility and thermal properties of the obtained copolymers. Polysulfones are particularly chemically resistant. This feature, and their good mechanical properties, resistance to X-rays and hydrolytically stability, them to have numerous applications (for example, in electrical devices, membranes for gases separation, containers for storing medicines, respirators, elements of machines and plumbing) [5,6,7,8]. Nowadays, the area of commercial use of polymeric materials is more and more demanding. Therefore, it is necessary to obtain materials that will satisfy the requirements of various industries. One good idea is the addition of new ingredients to these compounds to aim to obtain products based on already known substances. Because the properties of the polymers can be improved by the copolymerization of more monomers, so Joshi and Parsania synthesized the copolysulfonate of 1,1′-bis(3-methyl-4-hydroxyphenyl) cyclohexane, bisphenol A and 4,4′-diphenyl disulfonyl chloride. The researchers used an interfacial polycondensation technique at 5 °C and obtained a copolymer possessed of good hydrolytic stability against water, acids, alkali and salt solutions at 35 °C, good glass transition temperature and excellent thermal stability [9].

Macromolecular materials composed of at least two different chemicals can be divided into four groups: copolymers; polymer composites; polymer blends; and Interpenetrating Polymer Networks (IPNs) [10]. In the macromolecules of copolymers, there are at least two different types of repeated units. Their composition depends on the relative concentration of the monomers and their reactivity. These compounds can differ from each other in the arrangement of the repeated units (alternating, random, block, grafted ones) [11,12]. The polymer composites are the materials composed of a polymer matrix in which the filler is placed. Depending on the target use of the composite, an appropriate polymer or copolymer is selected for the composite matrix creation. The same refers to the filler, which can be a polymer, metal or ceramic material [13,14]. The characteristics that distinguish the composite from other multicomponent materials is a form of reinforcement (woven fabric, fiber, plate or rods) [15,16,17]. Due to the lower toxicity, biodegradability and sustainable development, the composites based on the natural polymers, such as cellulose or lignin, deserve special attention [18,19,20,21]. The polymer blend is a material, which is composed of several components and is a macroscopically homogeneous mixture, where the common forces between the components are only Van der Waals forces, or hydrogen bonds between the components [22,23]. In many cases they exhibit new, exceptional features, which are usually different compared with those of pure components [24,25]. The optimal properties of these materials result from selecting the appropriate proportions of the constituents: organic fillers and the polymer matrix in which the particles are dispersed. An important disadvantage of such materials is that the majority of the blends are heterogeneous mixtures showing a heterogeneous morphology [26,27]. In recent years, considerable progress has been observed in the field of polymer blends. The Figure 1 show the main distinguish features for composites, polymeric blends and copolymers.

At the beginning of the twenty-first century, polymer blends accounted for approximately 36% of the total production of polymers. The polymer blends have found application in both life, industry and advanced technologies [28,29]. Among the many special features that determine the popularity of the blends, the following should be mentioned:The possibility of obtaining materials with precisely designed properties at low cost;The possibility of modifying a selected feature;The increase in the efficiency of polymer production plants using the existing equipment as well as production lines for the production of new types of materials [24,30,31].

The aim of this study was to synthesize new polymeric blends based on ethoxylated bisphenol A (Ebecryl^®^ 150) and diacrylate ester of bisphenol A epoxy resin (Ebecryl^®^ 600) with the addition of NVP as an active diluent and different amounts of bisphenol Z polysulfone. Before the polymerization, the polysulfone was dissolved in NVP and then the UV polymerization reaction was performed. The characterization of the spectroscopic, thermal and thermo-mechanical properties is discussed in detail.

## 2. Materials and Methods

### 2.1. Materials

As the compounds for the synthesis of the bisphenol Z, the following were used: cyclohexanone (Sigma Aldrich, Darmstadt, Germany); phenol (POCh Gliwice, Poland) and hydrochloric acid (POCh Gliwice, Poland). The β-mercaptopropionic acid (used as the reaction catalyst) was delivered by Merck KGAA (Darmstadt, Germany).

As the compounds for the synthesis of the bisphenol Z polysulfone, the following were used: 4,4′-dichlorodiphenylsulfone (DCDPS) (Sigma Aldrich, Darmstadt, Germany) and 4,4′-cyclo-hexylidene bisphenol (bisphenol Z) which was synthesized in the Department of Polymer Chemistry, Maria Curie–Skłodowska University in Lublin, Poland. The potassium carbonate, methylene chloride, N, N-dimethyl-acetamide (DMAc) as a solvent and toluene (for formation azeotropic mixture with water) were delivered by Sigma Aldrich. The acetic acid and methanol were obtained from POCh Gliwice (Poland), and used for the precipitation of PSU-BPZ.

In the curing procedure of blends the following acrylic resins were used: Ebecryl^®^ 150 (EB-150) and Ebecryl^®^ 600 (EB-600) which were delivered by Allnex, The Coating Resins Company (Frankfurt, Germany). These substances were used as a main monomer, while N-vinyl-2-pyrrolidone (Fluka, Buchs, Switzerland) acted as second monomer and the reactive solvent of the reaction medium. Bisphenol Z polysulfone was used as a blends dopant. The 2,2-dimethoxy- 2-phenylacetophenone (Irqacure^®^ 651) was applied as the polymerization photoinitiator (Sigma Aldrich, Darmstadt, Germany).

### 2.2. Methods

The attenuated total reflection-Fourier transform infrared (ATR/FT-IR) spectrum of polysulfone was obtained using a Bruker FT-IR spectrophotometer TENSOR 27 with a diamond crystal (Bruker GmbH, Ettlingen, Germany). The spectrum was recorded in the frequency range of 3600–600 cm^−1^ with a resolution of 4 cm^−1^ and 64 scans in the transmittance mode, using the powdered sample of PSU-BPZ.

The ^1^HNMR spectroscopic analysis was performed in deuterated chloroform (CDCl_3_) determining the chemical shifts against the tetramethylsilane (TMS) standard, using a Bruker Avance 300MSL analyzer (Germany) at a resonance frequency of 300MHz and the number of scans was 2048.

The thermogravimetric analysis (TG) and the differential thermogravimetric analysis (DTG) were performed with the helium flow (25 cm^3^/min) at the heating rate 10 °C/min in the temperature range 25–600 °C with a Netzsch STA 449 F1 Jupiter thermal analyzer (Netzsch, Selb, Germany). The initial mass of the samples was ~5 mg. All of the TG measurements were taken in Al crucibles with pierced lids. An empty Al crucible was used as a reference.

The calorimetric measurements were conducted in the DSC Netzsch 204 calorimeter (Netzsch, Selb, Germany). The samples (~10 mg) were placed in the aluminum pans with pierced lids. An empty crucible was used as a reference. Dynamic scans were obtained at a heating rate of 10 °C/min in the temperature range from 25 °C to 600 °C, in the nitrogen atmosphere (flow rate: 30 cm^3^/min).

The dynamic mechanical analysis (DMA) was conducted using the DMA Q800 apparatus from TA Instruments (New Castle, NY, USA), provided with a dual-cantilever device. The thermo-mechanical properties of the cured materials were determined from the storage modulus (E’/20 °C) and the tension–loss tangent (tg δ), which were obtained as a function of the temperature. The experiments were conducted using the rectangular samples of the dimensions close to 2 ± 0.1 mm thick, 10 ± 0.1 mm wide and 65 ± 0.1 mm long. To this end, the rectangular bars of the samples (of the above dimensions) were prepared by the CNC-milling machine MFG 8037P (Ergwind, Gdańsk, Poland). The materials were longitudinally deformed by small sinusoidal distortion of 1 Hz frequency in the temperature range from −50 to 200 °C with a constant heating rate of 4 °C min^−1^. 

### 2.3. Synthesis of 4,4′-Cyclo-Hexylidene Bisphenol “Bisphenol Z”

The synthesis of bisphenol Z was performed in a mechanical stirrer, thermometer, reflux condenser and heating mantle. The amounts of the chemicals used in this preparation are shown in Table 1, whereas Figure 2 shows a simplified scheme of reaction.

Cyclohexanone, phenol and concentrated hydrochloric acid were placed in the flask. After stirring for 30 min (in room temperature), β-mercaptopropionic acid was added to the contents of the flask. The reaction was continued for another 2 h at a temperature of about 75 °C. The obtained product was filtered under vacuum and then washed ten times with hot distilled water to remove the excess phenol. The final product was crystallized from 96% ethanol.

### 2.4. Preparation of Bisphenol Z Polysulfone

The synthesis of polysulfone was performed in a four-necked round bottom flask, equipped with: a thermometer; a mechanical stirrer; an azeotropic head with a reflux condenser and a glass tube connected to the nitrogen gas. The amounts of reactants used in this synthesis are shown in Table 2.

Bisphenol Z, DCDPS, potassium carbonate, DMAc and anhydrous toluene were placed in the flask and, while stirring, the mixture was heated at 140 °C (for 8 h) with a nitrogen gas supply. After the distillation of the water formed during the reaction, toluene was also distilled from the reaction medium and the polymerization was carried on for 16 h at 160 °C. After the reaction was completed, methylene chloride was added to the mixture, and then the flask was heated to 40 °C. The flask contents were filtered under the reduced pressure and acetic acid was added to the filtrate. The polysulfone was precipitated using methanol and the next was dried in an electric dryer at a temperature of about 110 °C [32]. Figure 3 shows a simplified scheme of the PSU-BPZ polymerization process.

### 2.5. Preparation of Polymeric Blends

The process of the preparation of the samples included several stages. At the beginning, the calculated amounts of bisphenol Z polysulfone (PSU-BPZ) and NVP were placed in a three-necked round bottom flask. Next, the system was left for 24 h at room temperature to pre-dissolve the polysulfone in the solvent. After this time, the resin (EB-150 or EB-600) was added to the flask in accordance with the given weight proportions, then the flask was placed in a stand and equipped with a mechanical stirrer, thermometer, reflux condenser and heating mantle. The structures of the substances are shown in Figure 4, whereas Table 3 presents the proportions of components in each composition.

The amounts of the compounds in the mixtures are expressed in wt.% of the sample. With the continuous heating of the system at about 90 °C, the flask contents were stirred until a homogeneous form was obtained, then the photoinitiator (Irgacure^®^ 651) was added to the flask and further mixing was carried on until the photoinitiator was completely dissolved. Next, the mixture was poured between two glass plates 12 × 12 cm (in a form specially designed for this purpose) with a Teflon spacer (0.2 cm) and lubricated with silicon grease. This assembly was then placed in the UV chamber and the composition was cured for 2 min. After taking out the mold, the samples were ready for further analysis. In addition, using this method the polymer matrices i.e., NVP copolymers with the resins were synthesized to determine the changes in the selected properties. Figure 5 shows the samples of the obtained copolymers and blends. It can be seen that, with the change in the amount of resin in the composition, the change of color and transparency of blends was observed.

## 3. Results

### 3.1. ATR/FT-IR Analysis of Bisphenol Z Polysulfone

The chemical structure of the obtained bisphenol Z polysulfone was identified by the FT-IR spectroscopy. Figure 6 presents the FT-IR spectra of the PSU-BPZ. The characteristic bands of the methylene groups are visible as two signals: the first is at 2947 cm^−^^1^ and the second is at 2855 cm^−^^1^. These peaks correspond to the symmetrical and asymmetrical stretching vibrations of these groups presented in the cyclohexane ring. Additionally, the signals of the deformation vibrations of the methylene groups are notable: the signal at 1322 cm^−^^1^ is derived from twisting and wagging vibrations and the doublet: 869,831 cm^−^^1^ corresponds to the rocking vibrations of -CH_2_-. In the spectrum, the characteristic bands are derived from the benzene rings in the structure of the PSU-BPZ: 1584 cm^−^^1^ corresponding to the symmetrical vibrations of the aromatic rings and two signals: 1492 and 1478 cm^−^^1^ from the analogous asymmetric-stretching vibrations. The deformation vibrations of Ar and Ar-H are visible near 689 cm^−^^1^. The absorption due to the asymmetric-stress vibrations of the S = O bond in the -SO_2_- groups appears at 1295 cm^−^^1^, and that of the symmetric stress at 1102 cm^−^^1^ [33].

### 3.2. Proton Nuclear Magnetic Resonance Spectroscopy of Bisphenol Z Polysulfone

Figure 7 shows the ^1^HNMR spectrum of bisphenol Z polysulfone. Seven signals from protons in the PSU-BPZ molecule were observed:

In the range of chemical shifts δ: 6.93–7.85 ppm these are signals from hydrogen atoms attached to aromatic rings;In the range of chemical shifts δ: 1.52–2.26 ppm these are signals from the hydrogen atoms attached to the cycloalkane ring.

### 3.3. TG/DTG Analysis

The thermal stabilities and degradation behavior of the bisphenol Z polysulfone and polymeric composites were studied by means of thermogravimetry. The curves obtained from the TG and DTG (differential) measurements (in helium) for all if the samples are presented in Figure 8 and Figure 9. The main parameters of the thermal degradation: T_2%_-corresponding to the temperature of 2% of mass loss; T_50%_ (temperature at 50% weight loss of sample); maximum decomposition temperature (T_MAX_) and the residual mass (RM) at the final temperature for each sample, are listed in Table 4 and Table 5.

The materials were considered stable until 2% of their mass was lost. For the analyzed polysulfone, T_2%_ is 101.9 °C. On the TG and DTG curves, the small weight loss of the PSU-BPZ is visible in the range: 110–250 °C. This effect is not total degradation and rather derives from breaking the single bonds in the polysulfone structure; additionally, it may be related to the evaporation of small amount of adsorbed water. At 506.8 °C, an intensive peak derived from the main step of the thermal degradation of PSU-BPZ is observed on the DTG curve. This parameter, together with the value of the temperature at 50% weight loss (515.8 °C), show that the synthesized bisphenol Z polysulfone has great stability in high temperature conditions.

Comparing the course of the whole of the TG and DTG curves, it can be stated from the measurements that the copolymers behaved in a similar way for each sample. The initial decomposition temperatures (T_2%_) of the analyzed materials range from 72.1 to 86.1 °C, and it can be seen that the higher values of the temperature are related to the samples based on Ebecryl^®^ 150. For the analyzed copolymers, the TG and DTG curves had almost the same course, up to a temperature of ca. 300 °C. The main decomposition peak was observed in the range from 350 to 480 °C, with the maximum weight loss at 419.3 °C (33 wt.% of EB-600)–429.5 °C (67 wt.% of EB-150). This stage is connected with the total degradation of the polymeric matrices of the samples (destruction of aromatic fragments in the structure). Additionally, a very small decomposition signal in the wide temperature range (230–300 °C) for each material can be also found on the DTG curves. This stage is probably associated with the decomposition of the ester fragments derived from the aliphatic acrylates [34,35,36]. The analysis of the data presented in Table 4 shows that, in the case of both pairs of copolymers, the addition of a larger amount of resin to the samples increased its thermal stability to a similar extent. The largest thermal residue mass (RM) of copolymers with both of the resins determined at the final temperature was in the range from 1.73% (33 wt.% EB-150) to 6.3% (67 wt.% EB-600). It can be observed that the addition of a larger amount of EB-150 or EB-600 to the polymer network results in the increased RM in both pairs of materials.

According to the presented results, the blends with 10 wt.% bisphenol Z polysulfone were characterized by higher values of T_2%_ than those containing only 5 wt.% of the dopant. This is analogous to both of the used resins. Analyzing the DTG curves, one can see that, in the case of the blends, the curves contain one main signal related to the degradation stage and a small effect which ranged from about 180 to 300 °C. This signal (accompanied by a small loss of the sample mass) was assigned to the volatilization of small amounts of unreacted monomers and decomposition of fragments of resins (e.g., ester and ether groups) and polysulfones structure (e.g., sulfur dioxide). The maximum peak of the distribution (T_MAX_) refers to the total degradation of the analyzed materials. The range of this peak for the samples with 5 wt.% of PSU-BPZ takes place between from 424.1 °C (33 wt.% of EB-600) to 431.8 °C (67 wt.% of EB-150). The similar situation is visible In the case of the materials doped with 10 wt.%. of polysulfone. The range of this parameter is from 424.6 °C (33 wt.% of EB-600) to 433.8 °C (67 wt.% of EB-150). Compared to the samples without PSU-BPZ, the values of the individual parameters obtained thanks to the TG/DTG analysis are higher for the blends. Moreover increase amount of dopant cause the improvement of thermal resistance of polymer compositions.

As with the copolymer series, the data show that the residual mass increases with the increasing content of the resin used in the composition. Moreover, the presence of polysulfone in the polymer matrix resulted in the higher residual (final) weights of each sample of blend than the values of this parameter for the corresponding copolymers. The final masses of the blends containing EB-150 covered the range: 2.1–13.7%. The numerical data derived from the materials based on EB-600 show that these samples were characterized by a residual mass from 6.9% to 19.6%. The comparison of the above measurements shows (similarly to the copolymers) that the samples were characterized by a greater thermal resistance when they included 67 wt.% of EB-150 or EB-600 in their composition.

### 3.4. Differential Scanning Calorimetry of Materials

The glass transition temperature corresponds with the reversible transition in the amorphous regions of materials from a hard and relatively brittle “glassy” state into a molten (rubbery) state as an appropriate temperature is reached. As the PSU is not crystalline polymer, the DSC curve is expected to be quite flat, with the only change in the baseline being the glass transition temperature and degradation effect of the polymer. The thermogram of the heating during the DSC measurement of the PSU-BPH are shown in Figure 10.

The thermogram of the PSU-BPH shows a glass transition effect which appears at about 180 °C (glass transition temperature—Tg). The Tg is followed by the undisturbed course of the curve. No endo or exo effects are not observed until the intensive endothermic effect, whose maximum appears at 504 °C. This effect correspond to the thermal degradation of the polysulfone

The results of the DSC analysis, as curves for the materials obtained owing to the photopolymerization of the acrylic resins and the active solvent, are presented in Figure 11 and Figure 12.

As can be seen in Figure 11, the course of the DSC curves is similar for the pairs of the samples containing the same resins. The main endothermic effect for all of the copolymers and polymeric blend samples is close to 430 °C. For the samples without polysulfone, the range of this signal is from 427.6 °C to 432.4 °C. This effect is preceded by a smaller, also endothermic effect, the maximum values of which were read at much lower temperatures: for the samples with EB-150 this signal occurs at about 78 °C, whereas for the samples with EB-600, this small endo effect appeared near 130 °C. These additional effects on the DSC curves were most likely related to reaching the glass transition temperature. However, due to the wide and irregular shape, it was impossible to determine the exact value of this parameter, so it was determined using the dynamic mechanical analysis in the next stage of the experiment.

Each curve for the 5 wt.% or 10 wt.% of the PSU-BPZ containing the samples shows the endothermic effect associated with the total thermal degradation of the sample. The range of these signals for the E-150 based samples (containing 5 wt.% of dopant) was from 433.6 to 434.9 °C, while for the blends containing 10 wt.% of the PSU-BPZ was: 433.4–434.9 °C. On the other hand, for the materials based on EB-600, the ranges in which the peaks of the thermal degradation effects occur are: 426.5–426.1 °C (for 5 wt.% of dopant) and 431.4–433.5 °C (for the samples with 10 wt.% of polysulfone). Thus, it can be assumed that the type of acrylic resin could have a slight effect on the maximum value of the endothermic effect. Similarly to the previous results, the DSC curves for the blends show additional small endothermic effects occurring before the degradation process. For each curve, the signal had a wide and irregular shape, so the glass transition temperature was defined using the DMA analysis.

### 3.5. Dynamic Mechanical Analysis of Materials

After cutting out rectangular shapes by the CNC-milling machine, the cross-linked samples were subjected to the DMA analysis. Figure 13, Figure 14 and Figure 15 show the relationships between the storage modulus (E’) versus temperature and the mechanical loss factor (tg δ) versus temperature. It was impossible to determine the glass transition temperature using the DSC technique, because this process took place in a temperature range that was too wide. Therefore, the dynamic mechanical analysis was used for this purpose. In this paper, the position of the tan δ maximum was taken as the glass transition temperature (Tg) (associated with the process of segmental relaxation) [37] In Table 6 and Table 7 the results of the DMA analysis are shown.

Figure 13 shows the major changes in the storage modulus for the copolymers when the materials pass through the glassy state. The storage modulus changes in a similar way for each sample, but as a perfect plateau cannot be seen, it is difficult to describe the samples’ transition to a largely elastic state. The range of this parameter was from 3.17 to 3.24 and the sample: EB-150(50):NVP(50) was characterized by the greatest value. The selected results from the DMA analysis of the obtained copolymers are presented in Table 5. The width of the tan delta peak reflects the size of the sample heterogeneity. The narrow peak indicates its homogeneous structure [38,39]. On examining the courses of the tg δ curves, it can be seen that analyzing the tan delta values shows that the glass transition region spreads over a wide temperature range, regardless of the composite type and each curve exhibits two maxima which prove the large cross-linking and thus heterogeneity of the obtained materials. These transition regions show a similar degree of structural heterogeneity of the studied materials. The distance between the peaks is smaller for the samples containing 67 wt.% of acrylic resins. The values of tg δ max are from 0.28 to 0.34, whereas slightly larger values of this parameter are observed for the EB-150-containing copolymers.

In the storage modulus for the blends (Figure 14 and Figure 15), major changes are observed when the materials pass through the glassy state. The changes in the storage modulus for each sample is characterized by a different in the courses at the beginning of the glass transition region, which extends over a wide temperature range. E’/20 °C is highest for the compositions with 50 wt.% of EB-150 (or EB-600) for both 5 wt.% and 10 wt.% of the PSU-BPZ. The values of the storage modulus were in the range of 2.97–3.48. The tangent δ curves for the obtained blends showed a similar trend, depending on the measurement temperature as for the copolymers samples. As a result, all of the samples were characterized by a degree of heterogeneity, which was confirmed by the presence of two glassing temperatures for each sample. The curves of tg δ show smaller differences between the two glass transition temperatures in the samples containing greater amounts of acrylic resins (regardless of the type of resin) similarly to the copolymers. It was observed that the amount of polysulfone used as the dopant had a negligible effect on the values of the DMA results. The materials are characterized by very similar values of tg δ, which proves that the attenuation of each blend is comparable. Comparing the glass transition temperatures, it proves that their values are smaller than those of the samples without polysulfone. For the samples with 5 wt.% of PSU-BPH the range of Tg_1_: 72.9–81.7 °C and of Tg_2_: 105.7–130.8 °C. On the other hand, the Tg_1_ of samples with 10 wt.% of PSU-BPH is included between 71.9 and 83.2 °C, whereas the range of Tg_2_ is: 102.1–137.8 °C. This allows for the conclusion that the presence of the polysulfone in the polymeric matrix influences the differentiation in terms of the glass transition temperatures and sample flexibility.

## 4. Conclusions

The data obtained during the thermal analysis confirm that the amount of polysulfone as a dopant to a small extent influences the thermal stability of the samples. The TG analysis demonstrated that for the analyzed samples, the TG and DTG curves had a quite similar trajectory. The DTG curves contain one main separate degradation step in the range of the maximum: 419.3–429.5 °C for the copolymers and 424.1–429.1 °C for the blends. This stage is associated with the total degradation of the samples. Compared to the copolymers, the thermal resistance is higher for the blends. The obtained data show that, as in the case of the series of the materials without the dopant, the addition of a larger amount of resin to the blends increases their thermal stability in a similar way. As follows from the DSC analysis, the shape of the calorimetric curves for the blends and for the samples without polysulfone is similar, depending on the kind of resin. Each DSC curve shows the endothermic effect associated with the total thermal degradation of the sample. These signals were preceded by smaller, also endothermic effects observed at lower temperatures (having wide and irregular shapes). The discussed effects were probably related to the volatilization and decomposition of small amounts of monomers and the glass transition process. The additional properties of the obtained materials were determined by the DMA analysis. The glass transition region extends over a wide temperature range for each sample and their tg δ curves showed a similar trend: there can be seen two maxima corresponding to the two glass transition temperatures of the samples. These transition regions show some structural heterogeneity and prove high cross-linking of the copolymers and blends. For the samples containing 67 wt.% of acrylic resins, the decrease of distance between the peaks was observed, which indicates that these compositions were more homogeneous. The larger values of the E’/20 °C were found for the samples containing 50 wt.% of resin and 50 wt.% of NVP. After the addition of the PSU-BPH to the compositions, the values of this parameter increased. These results prove that the presence of polysulfone influences the changes in the glass transition temperatures and the flexibility of the obtained materials. The above results indicate that polysulfone can be used as a dopant for the synthesis of the thermal- and mechanical-resistant polymeric blends based on the acrylic resin. Based on the changes in the quantity of the PSU-BPH and the different mutual proportions of matrix components, it is possible to modify their features for prospective applications. These materials can be used for filling cavities or the repair of other damages in the polymer coatings, wood or other materials. Such blends can be used in optical fiber technology (e.g., optical fiber coatings) because they can be cured fast by UV-radiation while pulling the optical fiber from the glass preform.

## Figures and Tables

**Figure 1 polymers-14-03148-f001:**
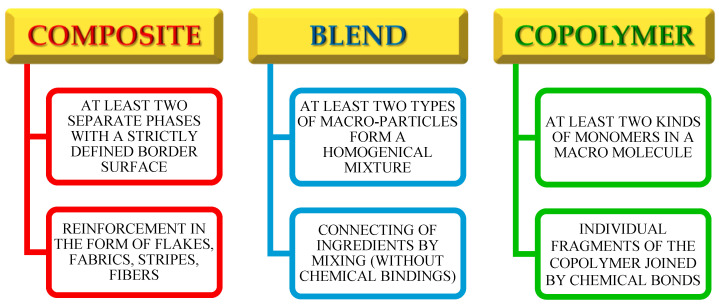
The most important features that distinguish composites, blends and copolymers.

**Figure 2 polymers-14-03148-f002:**
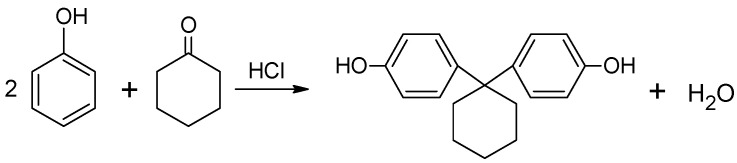
Reaction scheme for the synthesis of bisphenol Z.

**Figure 3 polymers-14-03148-f003:**
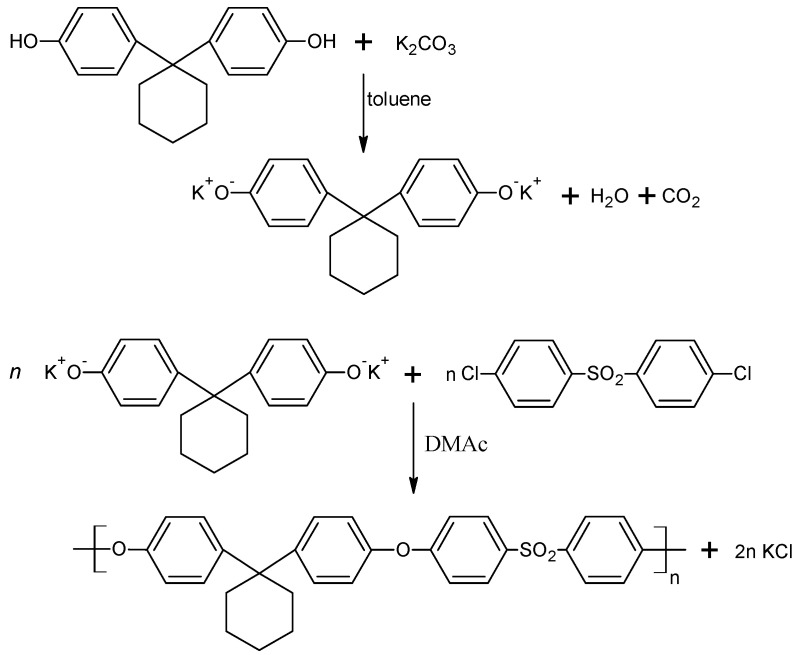
Reaction scheme for the synthesis of bisphenol Z polysulfone (PSU-BPZ).

**Figure 4 polymers-14-03148-f004:**
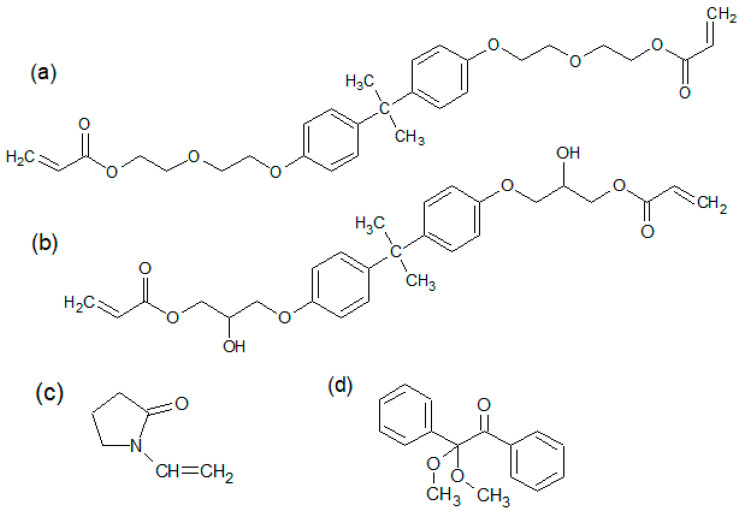
Chemical structures of the reactants used for preparation of copolymers and polymeric blends: (**a**) Ebecryl^®^ 150 (EB-150); (**b**) Ebecryl^®^ 600 (EB-600); (**c**) N-vinyl-2-pyrrolidone; (**d**) 2,2-dime- thoxy-2-phenyloacetophenone (Irgacure^®^ 651).

**Figure 5 polymers-14-03148-f005:**
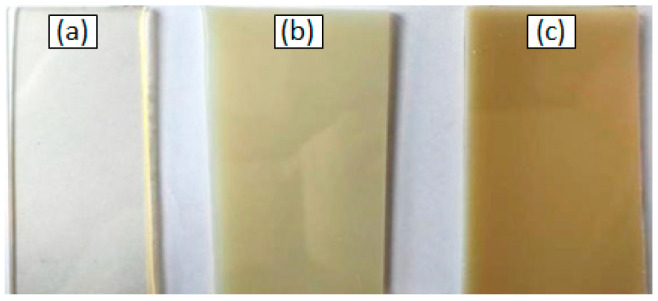
Color change and a decrease in transparency with an increase in the amount of polysulfone in the composition of the sample: (**a**) sample without dopant; (**b**) sample with 5 wt.% of PSU-BPZ; (**c**) sample with 5 wt.% of PSU-BPZ.

**Figure 6 polymers-14-03148-f006:**
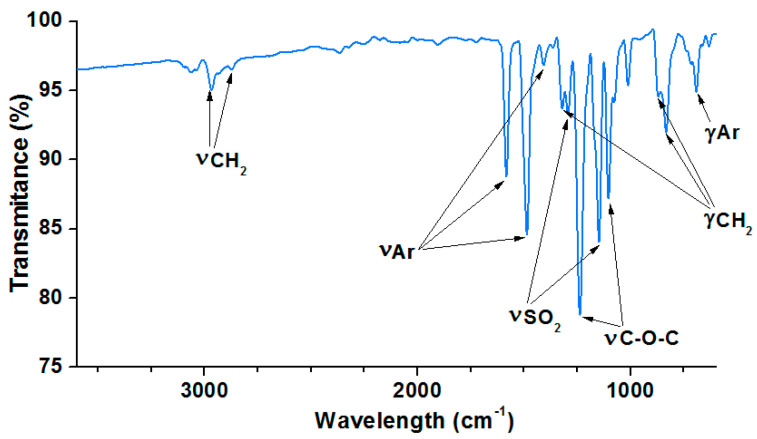
ATR/FT–IR spectra of the bisphenol Z polysulfone.

**Figure 7 polymers-14-03148-f007:**
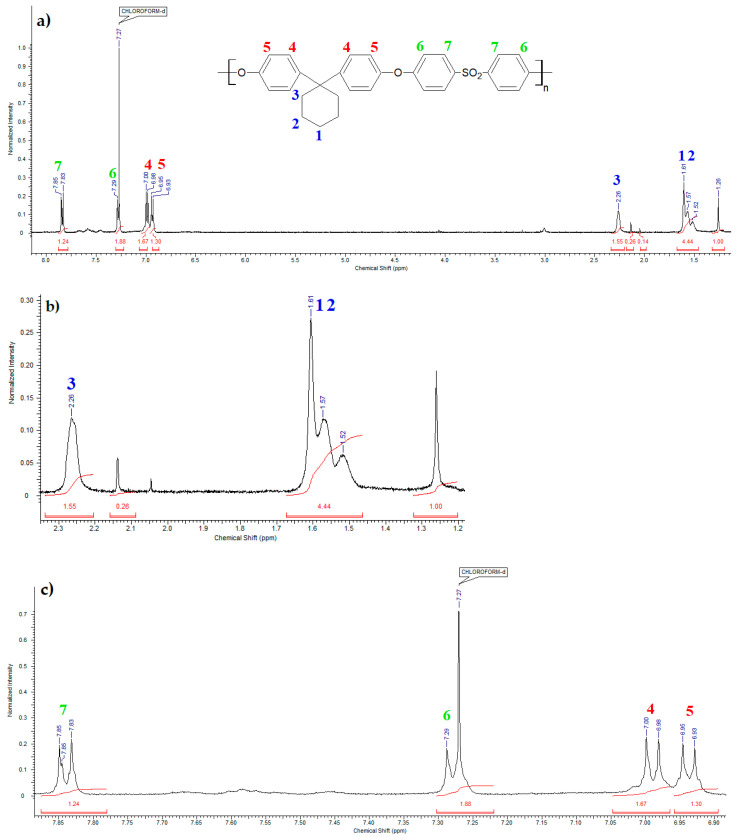
^1^HNMR spectra of the bisphenol Z polysulfone: (**a**) full spectrum range; (**b**) a fragment of the spectrum from 1.2 to 2.4 ppm (containing the signals of the cycloalcane rings); (**c**) a fragment of the spectrum from 6.9 to 7.9 ppm (containing signals from aromatic rings).

**Figure 8 polymers-14-03148-f008:**
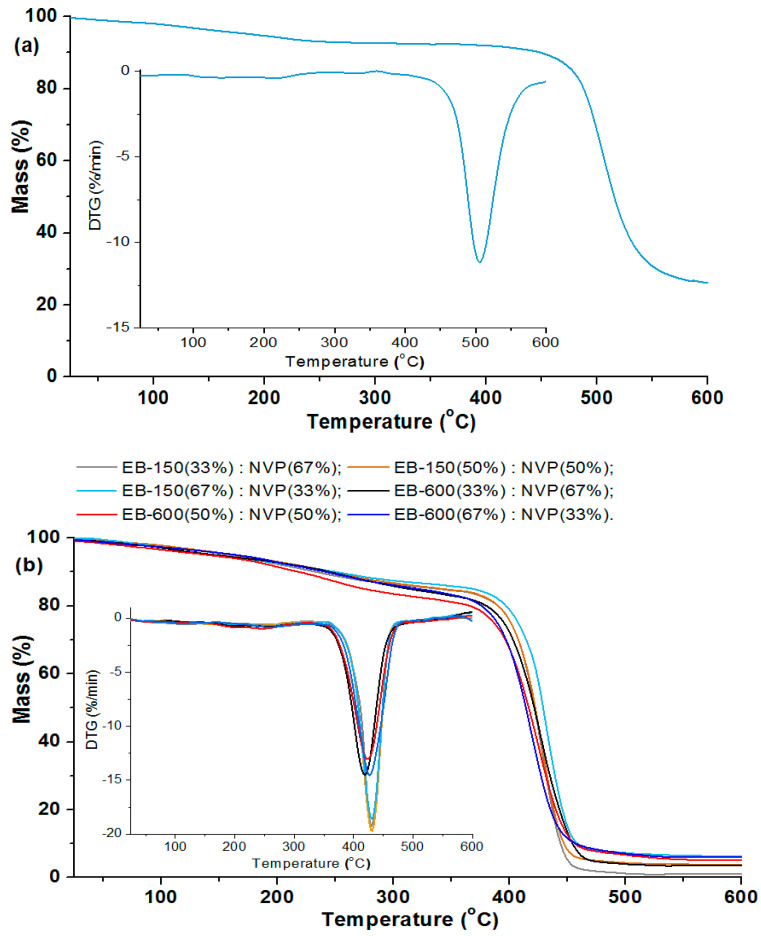
TG/DTG analysis of: (**a**) bisphenol Z polysulfone; (**b**) copolymers.

**Figure 9 polymers-14-03148-f009:**
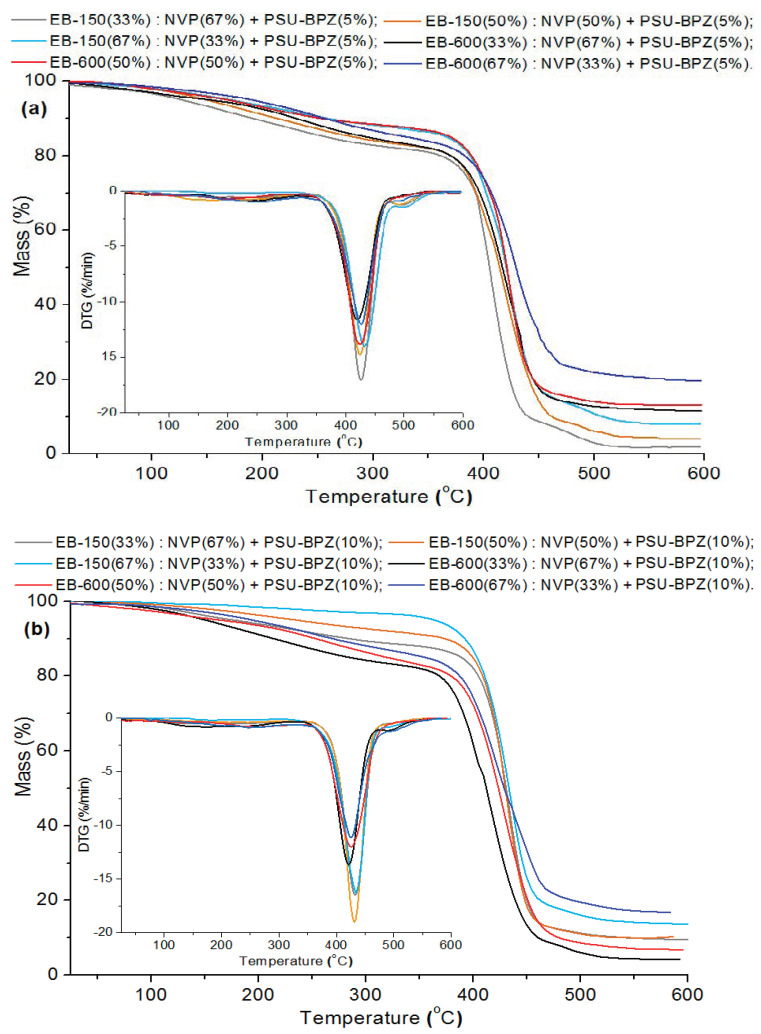
Thermal analysis: TG/DTG curves of blends: (**a**) for materials with 5 wt.% of PSU-BPZ; (**b**) for materials with 10 wt.% of PSU-BPZ.

**Figure 10 polymers-14-03148-f010:**
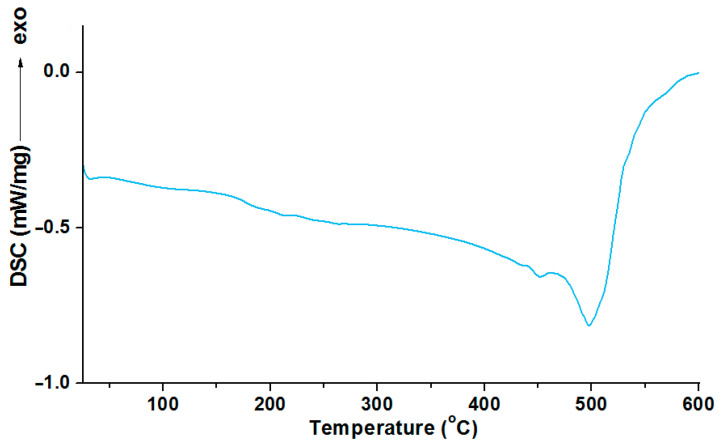
The curve of DSC analysis for bisphenol Z polysulfone.

**Figure 11 polymers-14-03148-f011:**
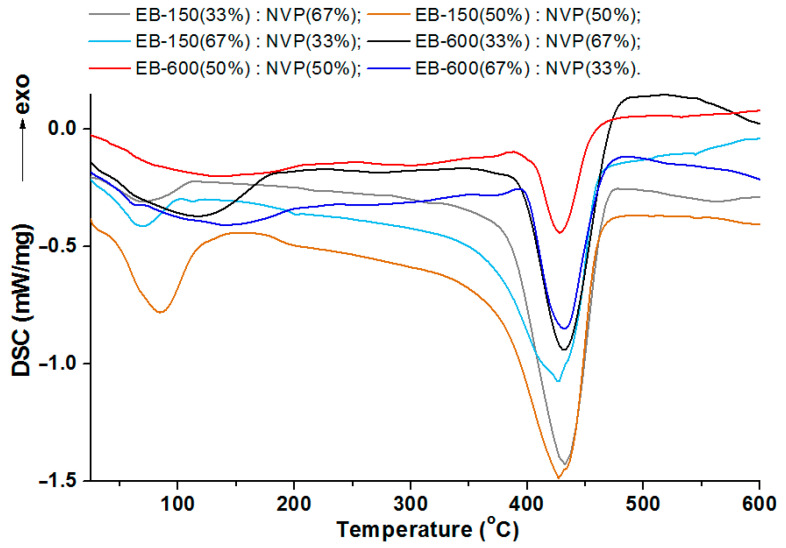
DSC curves for copolymers samples.

**Figure 12 polymers-14-03148-f012:**
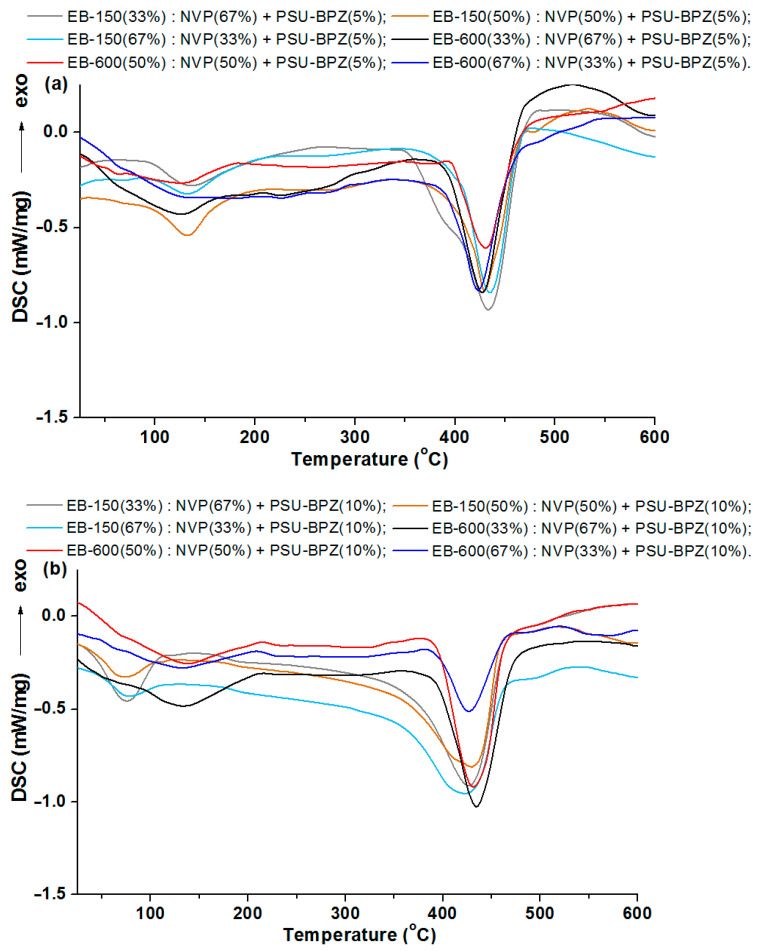
DSC curves of the blends: (**a**) samples with 5 wt.% of PSU-BPZ; (**b**) samples with 10 wt.% of PSU-BPZ.

**Figure 13 polymers-14-03148-f013:**
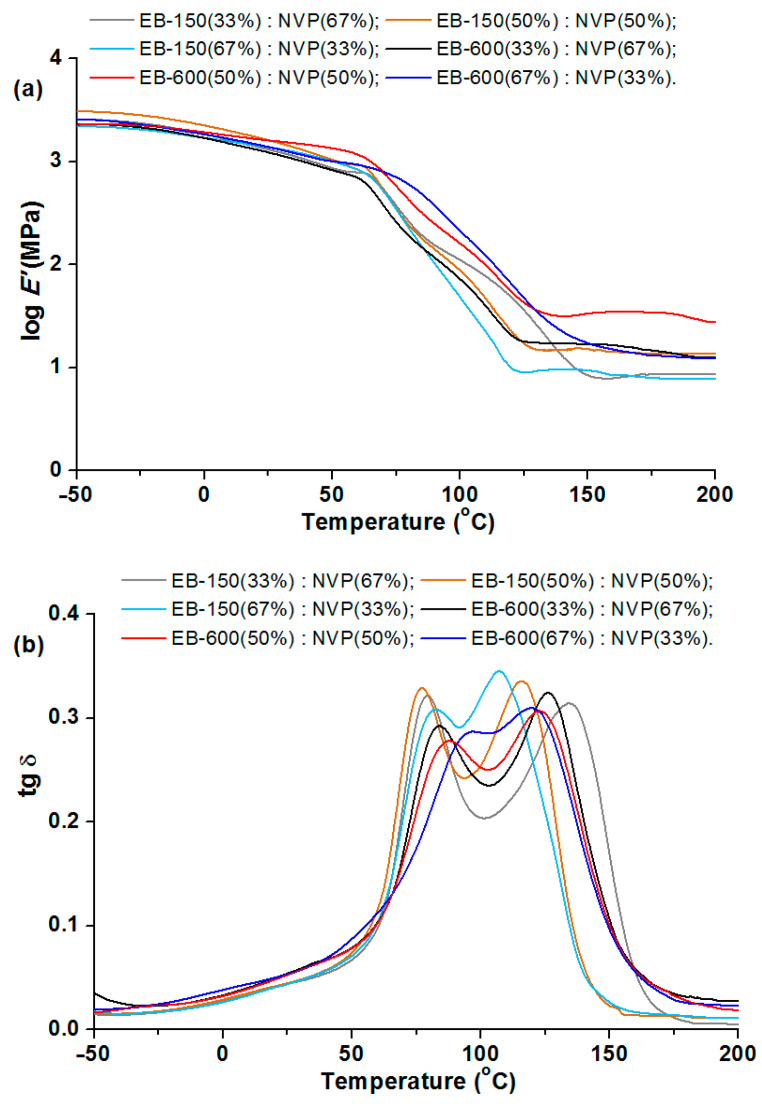
DMA analysis of copolymers: (**a**) storage modulus and (**b**) tangent delta in the function of temperature.

**Figure 14 polymers-14-03148-f014:**
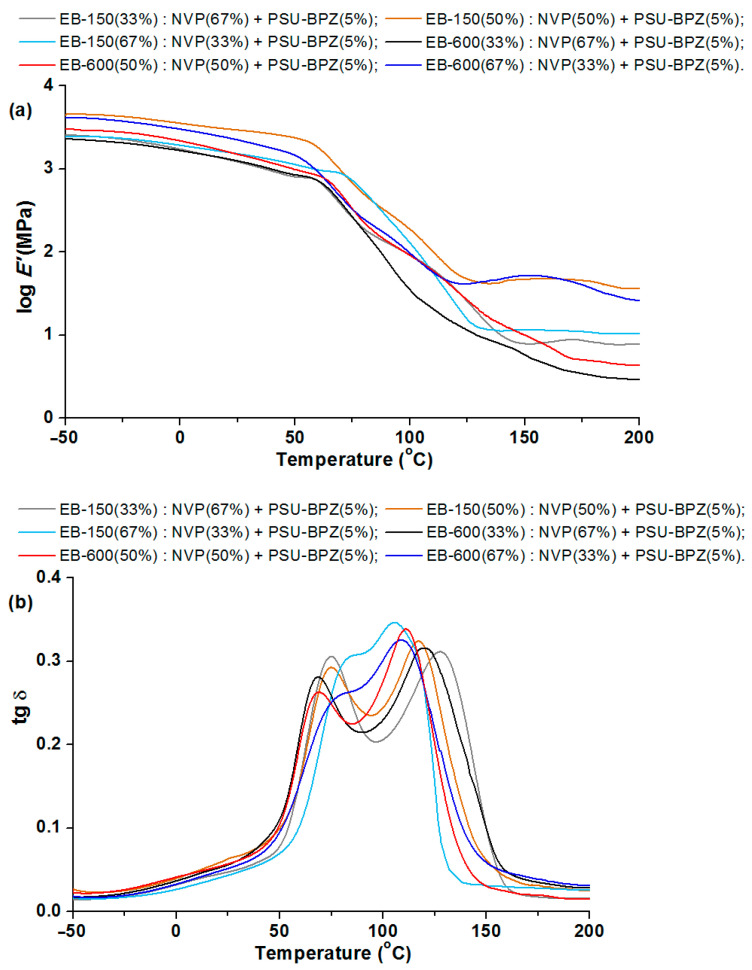
DMA analysis of polymeric blends with 5 wt.% of dopant: (**a**) storage modulus and (**b**) tangent delta in the function of temperature.

**Figure 15 polymers-14-03148-f015:**
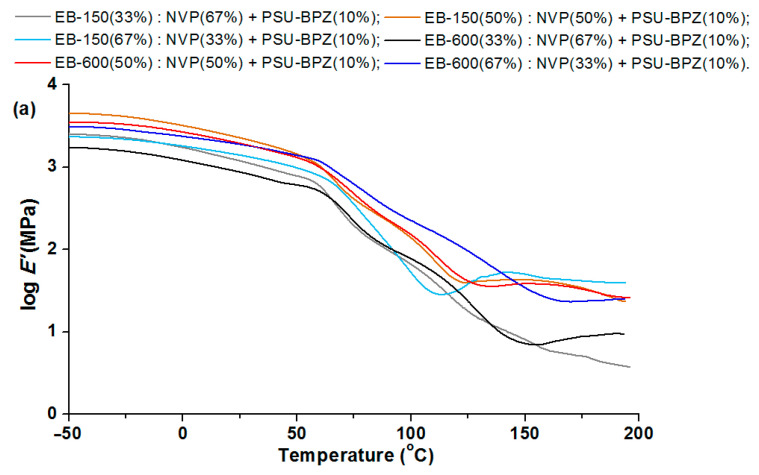
DMA analysis of polymeric blends with 10 wt.% of dopant: (**a**) storage modulus and (**b**) tangent delta in the function of temperature.

**Table 1 polymers-14-03148-t001:** Amounts of chemical compounds used for bisphenol Z preparation.

Name of Substance	Amount of Substance
cyclohexanone	98 g
phenol	376 g
hydrochloric acid	220 g
β-mercaptopropionic acid	3.66 g

**Table 2 polymers-14-03148-t002:** Amounts of chemical compounds used for bisphenol Z polysulfone polymerization.

Name of Substance	Amount of Substance
4,4′-cyclohexylidenebisphenol (Bisphenol Z)	16 g
4,4′-dichlorodiphenylsulfone (DCDPS)	17 g
potassium carbonate	9.6
N, N-dimethylacetamide (DMAc)	450 mL
toluene	200 mL
methylene chloride	330 mL
acetic acid	7.5 mL
methanol	1500 mL

**Table 3 polymers-14-03148-t003:** Amounts of chemical compounds used for polymerization of bisphenol Z polysulfone.

EB-150 or EB-600 (wt.%)	NVP (wt.%)	PSU-BPZ (wt.%)	Irgacure^®^ 651 (wt.%)
33	67	-	1
50	50	-	
67	33	-	
33	67	5	1
50	50	5	
67	33	5	
33	67	10	1
50	50	10	
67	33	10	

**Table 4 polymers-14-03148-t004:** Numerical values of the thermogravimetry analysis of the bisphenol Z polysulfone and copolymers without dopant.

	T_2%_ (°C)	T_MAX_ (°C)	T_50%_ (°C)	RM (%)
PSU-BPZ	101.9	506.8	515.3	26.1
EB-150 (33 wt.%)	81.3	427.6	418.8	1.1
NVP (67 wt.%)				
EB-150 (50 wt.%)	85.5	428.1	422.2	3.7
NVP (50 wt.%)				
EB-150 (67 wt.%)	86.1	429.5	428.3	6.3
NVP (33 wt.%)				
EB-600 (33 wt.%)	77.2	419.3	420.8	3.5
NVP (67 wt.%)				
EB-600 (50 wt.%)	72.1	423.5	416.6	5.2
NVP (50 wt.%)				
EB-600 (67 wt.%)	74.3	426.9	419.1	6.1
NVP (33 wt.%)				

**Table 5 polymers-14-03148-t005:** Data of thermogravimetric measurements of the blends containing of PSU-BPZ.

KERRYPNX		T_2%_ (°C)	T_MAX_ (°C)	T_50%_ (°C)	RM (%)
EB-150 (33 wt.%)		88.5	424.6	422.9	2.1
NVP (67 wt.%)	PSU-BPZ (5 wt.%)				
EB-150 (50 wt.%)		97.1	424.5	425.7	4.2
NVP (50 wt.%)					
EB-150 (67 wt.%)		105.6	431.8	427.1	8.1
NVP (33 wt.%)					
EB-600 (33 wt.%)		113.1	424.1	421.8	11.1
NVP (67 wt.%)					
EB-600 (50 wt.%)		120.4	425.2	421.1	13.2
NVP (50 wt.%)					
EB-600 (67 wt.%)		126.1	429.1	429.2	19.6
NVP (33 wt.%)					
EB-150 (33 wt.%)	PSU-BPZ(10 wt.%)	116.2	431.5	430.5	9.6
NVP (67 wt.%)					
EB-150 (50 wt.%)		148.6	430.9	429.8	10.3
NVP (50 wt.%)					
EB-150 (67 wt.%)		163.9	433.8	431.1	13.7
NVP (33 wt.%)					
EB-600 (33 wt.%)		120.1	424.6	412.1	6.9
NVP (67 wt.%)					
EB-600 (50 wt.%)		124.8	427.9	422.2	8.8
NVP (50 wt.%)					
EB-600 (67 wt.%)		130.6	429.9	428.7	16.6
NVP (33 wt.%)					

**Table 6 polymers-14-03148-t006:** Data of DMA measurements of copolymers.

	E’/20 °C (MPa)	Tg_1_ (°C)	Tg_2_ (°C)	Tg δmax_1_	Tg δmax_2_
EB-150 (33 wt.%)	3.14	78.7	134.2	0.32	0.31
NVP (67 wt.%)					
EB-150 (50 wt.%)	3.24	77.1	116.1	0.33	0.33
NVP (50 wt.%)					
EB-150 (67 wt.%)	3.16	82.6	101.6	0.30	0.34
NVP (33 wt.%)					
EB-600 (33 wt.%)	3.11	83.8	126.1	0.29	0.32
NVP (67 wt.%)					
EB-600 (50 wt.%)	3.22	87.6	123.4	0.28	0.30
NVP (50 wt.%)					
EB-600 (67 wt.%)	3.17	96.2	119.8	0.28	0.31
NVP (33 wt.%)					

**Table 7 polymers-14-03148-t007:** Data of DMA measurements of blends with 5 wt.% of PSU-BPZ.

		E’/20 °C (MPa)	Tg_1_ (°C)	Tg_2_ (°C)	Tg δmax_1_	Tg δmax_2_
EB-150 (33 wt.%)	PSU-BPZ (5 wt.%)	3.14	72.9	130.8	0.31	0.31
NVP (67 wt.%)						
EB-150 (50 wt.%)		3.48	72.8	117.1	0.29	0.32
NVP (50 wt.%)						
EB-150 (67 wt.%)		3.22	79.7	105.7	0.30	0.35
NVP (33 wt.%)						
EB-600 (33 wt.%)		3.12	78.4	120.5	0.28	0.31
NVP (67 wt.%)						
EB-600 (50 wt.%)		3.26	79.4	112.8	0.26	0.34
NVP (50 wt.%)						
EB-600 (67 wt.%)		3.21	81.7	114.5	0.26	0.32
NVP (33 wt.%)						
EB-150 (33 wt.%)	PSU-BPZ (10 wt.%)	3.16	71.9	127.9	0.31	0.35
NVP (67 wt.%)						
EB-150 (50 wt.%)		3.45	73.3	118.4	0.23	0.36
NVP (50 wt.%)						
EB-150 (67 wt.%)		3.23	82.2	102.1	0.29	0.38
NVP (33 wt.%)						
EB-600 (33 wt.%)		2.97	80.9	137.8	0.23	0.25
NVP (67 wt.%)						
EB-600 (50 wt.%)		3.32	81.9	129.9	0.28	0.35
NVP (50 wt.%)						
EB-600 (67 wt.%)		3.30	83.2	128.9	0.25	0.28
NVP (33 wt.%)						

## Data Availability

Not applicable.

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
