# Peer review of "Synthesis and Characterization of Polymeric Blends Containing Polysulfone Based on Cyclic Bisphenol"

_polymers, 2022, doi:10.3390/polym14153148_

Round 1

Reviewer 1 Report

The work has revealed the synthesis of polysulfone polymer and its impact on polymer blending.  The topic is very exciting and the study is highly informative based on its blending behavior. This work is of high interest to readers working in the field of the plastic industry.

The introduction part is more precise and reflects the work. The authors have shown the complete control experiments. 

Thus, it requires minor revisions in order to meet the journal's requirements.

1. Fig. 7 The NMR image should be improved thus readers can see the integration of the peaks. 

2. GPC of the synthesized polymers is missing. 

Author Response

Dear Reviewer,

Thank You for Your another review of our work, which will help with the elimination of discernible errors in our manuscript. We would like to express our gratitude for the reappearing revision of our work and the opportunity to re-submit it, incorporating all the suggestions. Our comments and changes were noted below, and  introduced in manuscript thanks using the “Track Changes” function.

Reviewer #1:

  1. Fig. 7 The NMR image should be improved thus readers can see the integration of the peaks.

Reply:

Thank you for this valuable suggestion. We have placed an two enlarged fragments of NMR spectrum of PSU-BPZ additionally.

  1. GPC of the synthesized polymers is missing.

Reply:

The standards used in GPC are dissolved in tetrahydrofuran (THF), so only this solvent was taken into account. However obtained polysulfone don't dissolved completely in THF (a cloudy heterogeneous solution formed), so in our opinion GPC analysis would be unreliable therefore it was not executed.

Reviewer 2 Report

I am encouraging and appreciate the authors to revise the manuscript according to the above suggestions that make it more interesting for the readers. And I think it could be published with major revisions.

Author Response

Dear Reviewer,

Thank You for Your another review of our work, which will help with the elimination of discernible errors in our manuscript. We would like to express our gratitude for the reappearing revision of our work and the opportunity to re-submit it, incorporating all the suggestions. Our comments and changes were noted below, and  introduced in manuscript thanks using the “Track Changes” function.

  1. Figures quality in figure 2 and 3 are very poor and need to redraw.

Reply:

We thank the Reviewer for this comment. The quality of the listed pictures has been improved.

  1. For NMR spectroscopic analysis, the author did not mention the number of scans for polymer structure analysis.

Reply:

We supplemented the description of the HNMR analysis with the number of scans.

Modification:

1HNMR spectroscopic analysis was performed (…) at a resonance frequency of 300MHz and the number of scans was 2048.

  1. What was the polymer concentration of the polymer solution when the sample was prepared for HNMR?

Reply:

Almost 8 mg of PSU-BPZ was dissolved in 0.8 ml of solvent. This gives a concentration of approx. 0.67%

  1. How is the differential scanning calorimetry (DSC) instrument calibrated for thermal analysis?

Reply:

The sensitive calibration and temperature calibration of calorimeter was calibrated at based on indium sample.

This manuscript is a resubmission of an earlier submission. The following is a list of the peer review reports and author responses from that submission.

Round 1

Reviewer 1 Report

The manuscript entitled “Synthesis and characterization of polymeric blends containing of polysulfone based on cyclic bisphenol” presents the synthesis of bisphenol Z polysulfones and its reactive blending with two acrylic monomers affording the respective resins. The influence of the blend’s composition on their thermal and mechanical properties, is examined.

However, the synthetic part of the manuscript contains the preparation of bisphenol Z and its polymerization resulting in the respective polysulfone. However, several bishenol Z polyether sulfones have been previously published even before 1990. So, all the discussion on characterization of the bisphenol Z polysulfones should be related with the previously published polymers and copolymers providing the respective references, and also make clear what is the new finding here that deserve publication. The comparison of the thermal and mechanical properties between the resins and the new blends shows that there are only marginal differences and thus there are no concrete conclusions of severe properties enhancement or identification of ultimate compositions that can be used in specific niche or industrial applications.

Finally, the introduction part is too long containing information not related to the specific polymers used in the study.

Reviewer 2 Report

Dear authors,

it is a good paper.

I have three comments:

1) There is an error in figure 1. You write "meres" (green box). Please change to monomers.

2) You write Bisphenol Z was synthesized at your institute. However, there is no reference to the literature. If it has not yet been published, then the synthetic route must be described.

3) For all thermal properties, the legend of the second figure is wrong. If you are looking at blends, please add PSU to the legend.

BR
The reviewer

Reviewer 3 Report

Mateusz Gargol et al. reported synthesis and characterization of polymeric blends containing 2 of polysulfone based on cyclic bisphenol. The work has revealed the synthesis of polysulfone polymer and its impact on polymer blending.  The topic is very exciting and the study is highly informative based on its blending behavior. This work is of high interest to readers working in the field of the plastic industry.

However, the work has significant methodological flaws to publish in this article. Thus, it requires major revisions in order to meet the journal's requirements.

  1. The characterization of polysulfone is missing. The authors had not presented NMR, GPC, TGA, DSC, etc. for the simple characterization. The authors have synthesized a new polymer it is necessary to have a scientific analysis of it.  
  2. The controlled experiments are missing. The authors have prepared copolymers based on 2:1 (resins and NVP) ratio. It is not clear why the authors have chosen this particular ratio? I would like to see a copolymer based on 1:1 and compare the data with other copolymers.   
  3. The authors have blended 5% polysulfone polymer with those copolymers. Few more blended copolymers with 10 or 20 percent polysulfone were necessary to have a scientific conclusion.